# *FGF22* Secreted by Hair Papilla Cells Regulates Hair Follicle Stem Cell Proliferation and Differentiation

**DOI:** 10.3390/biom15111560

**Published:** 2025-11-06

**Authors:** Yu Luo, Tong Xiao, Binpeng Xi, Yufang Song, Zengkui Lu, Chao Yuan, Jianbin Liu, Tingting Guo

**Affiliations:** 1Key Laboratory of Animal Genetics and Breeding on the Tibetan Plateau, Ministry of Agriculture and Rural Affairs, Lanzhou Institute of Husbandry and Pharmaceutical Sciences, Chinese Academy of Agricultural Sciences, Lanzhou 730050, China; 17393144478@163.com (Y.L.); xiaotong3110134994@163.com (T.X.); w5562080w@163.com (B.X.); songyf805@163.com (Y.S.);; 2Sheep Breeding Engineering Technology Research Center of Chinese Academy of Agricultural Sciences, Lanzhou 730050, China

**Keywords:** *FGF22*, hair follicle, dermal papilla cells, hair follicle stem cells, co-culture

## Abstract

Hair follicle stem cells (HFSCs) are resident stem cells within hair follicles (HFs) that possess self-renewal and differentiation capacities, serving as a critical model for regenerative medicine research. Their dynamic interaction with dermal papilla cells (DPCs) plays a decisive role in HF development and cycling. *FGF22* is a paracrine fibroblast growth factor that can regulate the proliferation, differentiation and migration of epithelial cells. This study established a DPC-HFSC co-culture system, revealing that *FGF22* overexpression in DPCs significantly upregulated *FGFR1*/*FGFR2* mRNA expression levels in HFSCs (*p* < 0.05), with a 1.67-fold increase in EdU-positive cell proportion (*p* < 0.01). CCK-8 assays demonstrated markedly enhanced HFSC viability (*p* < 0.01), with a 17% reduction in HFSC apoptosis (*p* < 0.05). Conversely, *FGF22* knockout downregulated *FGFR1/FGFR2* expression (*p* < 0.05), reduced HFSC proliferation capacity by 25% (*p* < 0.01), and increased HFSC apoptosis levels by 1.81-fold (*p* < 0.05). In addition, *FGF22* overexpression promotes the proliferation and differentiation of HFSCs by activating Wnt/β-Catenin, Sonic Hedgehog (Shh) and Notch signaling pathways, or inhibiting BMP signaling pathways. Knockout of *FGF22* weakens these processes and inhibits the activation and differentiation of HFSCs. This study, through the DPCs-HFSCs co-culture system, revealed the regulatory mechanism of *FGF22* secreted by DPCs on the proliferation and differentiation of HFSCs, thereby providing theoretical references for fields such as fine-wool sheep breeding, human regenerative medicine, and hair loss treatment.

## 1. Introduction

Hair serves as a pivotal structure in both animals and humans, fulfilling defensive, protective, communicative, and esthetic roles [1,2]. Hair follicles (HFs), the core component of hair, exhibit a cyclical regenerative ability and a high level of self—renewal, thereby rendering them an exemplary model for investigating cell proliferation, differentiation, and senescence [3]. Positioned in the follicular bulge region, HFSCs act as the fundamental unit driving the dynamic transition of the hair cycle, encompassing anagen, regression, and resting phases [4]. The activity of HFSCs is regulated by signaling pathways such as Wnt/β-Catenin, Shh, FGF, Notch, and BMP, which are secreted by DPCs located at the base of the follicle [5]. These signals originate from a wide range of sources, encompassing the HFSCs themselves, the DPCs, and exogenous cell types (e.g., adrenocortical cells) [6]. As pluripotent progenitors with remarkable self-renewal capacity, the HFSC possess the potential to differentiate into all cell lineages of the epidermis [7]. The proliferation and differentiation of the HFSCs are finely tuned by a signaling network that is centered on the DPCs [8]. Although DPCs play a key role in HFs development, there is still a lack of insight into the specific molecular processes and regulatory steps involved in their signaling to HFSCs. Therefore, an in-depth study of the interaction between DPCs and HFSCs is important for a comprehensive understanding of the mechanism of HFs development. Existing studies have confirmed that the morphogenesis and regeneration process of HFs is synergistically regulated by multiple signaling pathways such as Wnt/β-Catenin, Shh, Notch, BMP, and FGF [9]. Among them, the Wnt/β-Catenin signaling pathway plays a central role in hair regeneration and is able to activate HFSCs to initiate the hair growth process [10]. The Shh pathway drives HFs morphogenesis by inducing the proliferation of quiescent HFSCs and the Shh protein secreted by HFSC is an important initiator of HF regeneration in adults [11]. The Notch pathway promotes the activation of HFSCs in a T-reg-like manner [12]. Whereas the BMP pathway normally inhibits HFSCs activity, its antagonist *TGF-β2* promotes HFs regeneration by blocking it [13]. Above pathways regulate the cyclic regeneration of HFs through a dynamic balance.

FGF family members play a key role in regulating cell proliferation, differentiation, migration and survival [14]. Among them, the *FGF7* subfamily (*FGF3*/*7*/*10*/*22*) acts as specific activators of *FGFR1*/*2* and may have evolved from common ancestral genes [15,16]. Existing studies have revealed diverse functions of members of this subfamily in hair follicle biology. *FGF22* (Fibroblast growth factor 22), as a distinctive member of the FGF family, functions in the skin tissue. Relevant studies have shown that in the FGF family, *FGF5* mRNA is specifically expressed in the outer root sheath of hair follicles, while *FGF7* mRNA is expressed in the dermis [17]. When the *FGF5* gene was knocked out, the hair of mice showed abnormal growth [18]; The hair of *FGF7* gene knockout mice showed greasy or messy and fluffy characteristics [19]. These research results suggest that both *FGF5* and *FGF7* are involved in the hair development process. In contrast, *FGF22* mRNA is mainly expressed in the root sheath within hair follicles. Based on this, *FGF22* has the potential to become a special FGF involved in hair development. Co-culture with DPC-HFSC revealed that *FGF7* synergistically promotes the proliferation of both types of cells and induces HFSC differentiation through the Wnt pathway [20]. *FGF22* is involved in HFs cycle transition and induces it to enter into the full-growth phase earlier than expected [21]. *FGF5* knockdown increases the number of secondary hair follicles (SFs) in goats and *FGF18* knockdown shortens HFs resting phase in mice [22,23]. *FGF20* knockdown in fine wool sheep DPC impedes HFSCs differentiation by inhibiting Notch and Wnt/β-Catenin pathways [24]. Notably, *FGF22* is specifically expressed in dermal and follicular cells to regulate hair growth and skin regeneration [25]. Its developmental expression profile suggests that it is widely distributed in HFs structures and play a role in morphogenesis, as well as participating in inducing regression [26]. Although the available evidence suggests that *FGF22* is a key factor in the regulation of HFs growth, the mechanism of the DPC-HFSC communication mediated through the FGFR receptor remains unclear.

Based on this, this study focused on the regulation of *FGF22* secreted by DPCs on the proliferation and differentiation of HFSCs in fine-wool sheep. By constructing the *FGF22* overexpression/knockdown DPC model, and utilizing DPC-HFSC co-culture and recombinant *FGF22* stimulation experiments, we systematically investigated its influence on HFSCs biological behavior and signaling mechanisms. The aim was to provide new perspective into DPC-HFSC interaction network and to offer references for the development of the fine wool sheep industry.

## 2. Materials and Methods

### 2.1. Ethical Statement

All operations involving animals were carried out in strict compliance with the requirements of animal ethical procedures and norms in the People’s Republic of China. This study received approval from the Animal Management and Ethics Committee of the Lanzhou Institute of Animal Husbandry and Veterinary Medicine, Chinese Academy of Agricultural Sciences (Approval No. 023 1447. Approval date: 19 November 2023.).

### 2.2. Animals and Sample Preparation

We selected two 7-month-old healthy alpine Merino sheep from the Gansu Province Sheep Breeding Technology Promotion Station (Zhangye, China) and fed them the same diet. Select a location one palm’s width from the posterior edge of the scapula on the side of the body and use curved scissors to remove the wool. Then wipe the area with 75% alcohol to disinfect it, and use a sterile Acu-Punch skin biopsy punch (10 mm, Acuderm Inc., Norwalk, CA, USA) to collect a skin samples of approximately 1 cm^2^. Rinse the obtained samples twice with 75% alcohol, followed by three washes with PBS supplemented with 1% penicillin-streptomycin. Finally, store the samples in DMEM/F12 medium containing 1% penicillin-streptomycin, and transport them back to the laboratory for subsequent cell culture procedures.

### 2.3. Isolation, Purification and Identification of Primary DPCs and HFSCs

Under aseptic conditions, skin samples were sequentially immersed in 75% alcohol, 10× PBS and 1× PBS for 4–5 min each. Broken hair debris and adipose tissues were scraped using a sterile scalpel blade (Swann-Morton, Sheffield, UK) and 1 mm^3^ sized blocks of tissues were cut and incubated in 0.25% trypsin at 37° for 2 h. For isolation of HFSCs, intact single follicles were extracted from the skin tissues using a 1 mL syringe and placed in a 24-well plate for incubation. HFSCs adherent growth was observed after about 5–7 d. For the isolation of DPCs, a single HF was isolated, the tip of its hair bulb was scratched to extrude the hair papilla, and then was transferred to a 48-well plate for culture with the help of a mouth pipette, and the wall-adherence of the hair papilla cells was seen in about 12 d. According to the differences in the adhesion characteristics, DPCs and HFSCs with a purity of over 95% were obtained after three successive purifications with 0.25% trypsin.

Cellular immunofluorescence was employed for the identification of DPCs and HFSCs. DPCs were labeled using the proteins α-smooth muscle actin (α-SMA) and SRY-box transcription factor 2 (SOX2), while HFSCs were labeled with cytokeratin 14 (CK14) and cytokeratin 19 (CK19). DPCs and HFSCs at the fourth or passage 4 or passage 5 (P4 or P5) were chosen and cultured by inoculating them into 6-well plates. Once the cells reached 60–70% confluence, the culture medium was removed, and the cells were washed three times with PBS. Subsequently, the cells were fixed in 4% paraformaldehyde for 20 min, followed by another three washes with PBS. Next, the cells were incubated in 0.5% Triton X-100 (Beyotime, Shanghai, China) for 10 min, washed three times with PBS, and then blocked in 3% bovine serum albumin (BSA) for 1 h. After another round of three PBS washes, the cells were incubated with a 1:100 dilution of the specific primary antibody corresponding to the labeling protein at 4 °C. Similarly, the cells were washed three times with PBS and then incubated with a 1:100 dilution of Cy3-labeled goat anti-rabbit secondary antibody for 1 h at room temperature. Finally, the cells were washed three times with PBS, incubated with 4′,6-diamidino-2-phenylindole (DAPI) for 10 min at room temperature, and then imaged using a confocal microscope. Cell culture operations were carried out in an incubator with a constant temperature of 37 °C, a 5% CO_2_ concentration and humidification function. The specific medium conditions are shown in Table 1.

### 2.4. Overexpression and Knockdown of FGF22

The predicted mRNA sequence of the full-length CDS sequence of *FGF22* in fine wool sheep (543 bp, GenBank: XM_060416424.1) was provided by NCBI, which was directionally cloned into the pcDNA3.1 eukaryotic expression vector by double digestion with *Hind*III and *EcoR*I to construct the recombinant plasmid pcDNA3.1-FGF22. The negative control vector pcDNA3.1-NC, the recombinant plasmid pcDNA3.1-FGF22, as well as siRNA-FGF22 were all designed and synthesized by Gemma Gene Company (GenePharma, Suzhou, China) (Table 2).

### 2.5. DPC Transfection and Efficiency Test

The transfection efficiency of *FGF22* siRNA was determined prior to the construction of co-culture cell models. The passage 5 DPCs were inoculated into the lower chamber. Upon attaining 60–70% confluence, the DPCs were transfected with siRNA-FGF22-125, siRNA-FGF22-211, siRNA-FGF22-311, siRNA-FGF22-482, siRNA-NC, or FAM-labeled NC (designed and synthesized by Gemma Gene Company; sequence information is shown in Table 2). The transfection was performed using transfection reagents (Zeta Life, Inc., Menlo Park, CA, USA) in strict accordance with the manufacturer’s protocol, with a transfection reagent-to-siRNA ratio of 3:1. The efficiency of RNA interference-induced overexpression was quantified by qRT-PCR. The most effective siRNA-FGF22 was then identified for use in subsequent cell co-culture experiments.

### 2.6. Construction of DPC and HFSC Co-Culture Model

A co-culture model integrating DPCs and HFSCs was constructed with a Transwell cell chamber that has a 0.4 μm pore diameter (Figure 1). The Passage 5 (P5) DPCs were inoculated into the lower compartment of the chamber. Upon achieving 60–70% confluence, siRNA-FGF22 and siRNA-NC were introduced into the DPCs through transfection using transfection reagents. After a 24 h transfection incubation, Passage 5 (P5) HFSCs were added to the upper compartment and co-cultured with the DPCs in the lower compartment for a 24 h duration. Thereafter, a subset of the cells was collected for RNA isolation, and the mRNA levels were measured using qRT-PCR.

### 2.7. Total RNA Extraction and qRT-PCR

The extraction of cells from the co-culture model consisting of DPCs and HFSCs was carried out utilizing Trizol reagent (Takara Bio, Inc., Kyoto, Japan). Then RNA concentration and quality were assessed using the UV microspectrophotometer and electrophoresis, respectively. qRT-PCR primers for *FGF22*, its receptors *FGFR1* and *FGFR2*, as well as cell differentiation and signaling pathways, such as β-Catenin, *SOX9*, *c-myc*, *APC*, *Shh*, *GLI1*, *LGR5*, *Smo*, *NOTCH1*, *NOTCH3*, *HES1*, *DLL1*, *BMP2*, *BMP4*, *SMAD6* and *ID1*, were designed using Primer Premier 5.0 (Premier Biosoft International, Alto, CA, USA) software based on the mRNA sequences of the candidate genes published by NCBI (GenBank), and synthesized by Sangon Biotech Company (Shanghai, China) (primer sequences are shown in Table 3). The cDNA was synthesized using a PrimeScript RT Reagent kit containing gDNA Eraser (Vazyme, Nanjing, China). The cDNA was used as a template for qRT-PCR reactions using the ChamQ Universal SYBR qPCR Master Mix (Vazyme, Nanjing, China) on a CFX96 Touch Real-Time PCR Detection System (Bio-Rad Laboratories Inc., Hercules, CA, USA). The following components were used for qRT-PCR system (total volume 20 μL): 10 μL qPCR mix, 0.8 μL upstream and downstream primers, 2 μL cDNA (100 ng), and ddH_2_O. The cycling conditions for qRT-PCR are: pre-denaturation at 95 °C for 30 s; 40 cycles of 95 °C for 5 s (denaturation) → 60 °C for 10 s (annealing) → 72 °C for 30 s (extension); Melting curve analysis: 65–95 °C, heating rate 0.5 °C/s. Experiments were performed with the β-actin (*ACTB*) as the internal reference gene normalized to the cDNA template.

### 2.8. Proliferation and Viability Assessment in the DPC and HFSC Co-Culture Model

Following 24 h of co-cultivation, cell proliferation was assessed utilizing an EdU kit (Beyotime, Shanghai, China). The stock solution of EdU was diluted with cell culture medium in a 1:500 ratio. Subsequently, an equal volume of the diluted solution was added to the co-cultured cell plates in a 1:1 proportion, and the plates were incubated for 2 h. After discarding the medium, the cells were fixed with 4% paraformaldehyde at room temperature for 15 min and washed with 3% BSA-PBS for three times (each time for 5 min). The cells were treated with 0.3% Triton X-100 permeabilising solution for 15 min, washed again, and incubated for 30 min with the addition of click reaction solution (500 μL/well) in the dark. After three washes, the nuclei were stained with Hoechst 33342 (500 μL/well) for 10 min, and finally photographed for observation by fluorescence microscope. The CCK-8 kit (Beyotime, Shanghai, China) was employed to assess the cell viability of co-cultured DPCs and HFSCs. Specifically, 100 μL of CCK-8 solution was introduced into the upper chamber, while 200 μL was added to the lower chamber. The absorbance at 450 nm was then measured at 12, 24, 36, and 48 h intervals during the incubation period. To guarantee biological replicability, all experiments were carried out in triplicate.

### 2.9. Flow Cytometry Analysis of Apoptosis

Flow cytometry was utilized to detect apoptosis, employing the Annexin V-FITC apoptosis detection kit (Beyotime, Shanghai, China) and following the manufacturer’s guidelines. Passage 5 DPCs were seeded into 6-well plates and transfected with either the overexpression system (pcDNA3.1-FGF22 and pcDNA3.1-NC) and the knockout system (siRNA-FGF22 and siRNA-NC). Following a 24 h incubation period, the Passage 5 HFSCs were seeded into the upper chamber and allowed to co-culture for another 24 h. The cell culture medium was then carefully aspirated into a centrifuge tube. The adherent cells were rinsed once with PBS. Next, trypsin digestion was halted by adding the original culture medium, and the cells were resuspended through gentle pipetting. The resulting cell suspension was transferred to a fresh centrifuge tube and centrifuged at 1500× *g* for 5 min. After that, the supernatant was removed, and the cells were resuspended in PBS and centrifuged once more. Once the supernatant was discarded again, 195 μL of Annexin V-FITC binding solution was added to the cell pellet. Subsequently, 5 μL of Annexin V-FITC and 10 μL of propidium iodide (PI) staining solution were added in sequence. The cells were thoroughly mixed and then incubated at room temperature in the dark for 15 min. Afterward, the cells were temporarily placed in an ice bath. The flow-through assay was required to be completed within 1 h. The entire experiment was replicated three times.

### 2.10. Statistical Analysis

In this study, the 2^−ΔΔCt^ method of RT-PCR results was used for relative quantitative analysis, and the data were expressed as x ± s (mean ± standard deviation). Each sample was subjected to three technical replicates, with the experiment independently repeated three times (*n* = 3). Biological replicates were derived from distinct individual samples (e.g., obtained from independent cell passages or isolated from separate animal individuals). Statistical analysis was performed using SPSS 22.0 software for one-way analysis of variance (ANOVA). After multiple comparisons and corrections by the Tukey method, the *p* values were marked. Among them, *p* < 0.05 was considered a significant difference, and *p* < 0.01 was considered an extremely significant difference.

## 3. Results

### 3.1. Morphology and Identification of the Primary DPCs and HFSCs

Under aseptic conditions, primary DPCs were extracted from the skin tissues of fine-wool sheep via a combination of enzymatic digestion and mechanical separation techniques (Figure 2A). When observed under an inverted microscope at the 6th day of DPCs cultured, the cells grew adherently to the wall with the morphology of a pike, triangle, or polygon and a relatively large cell body (Figure 2A), which is consistent with that has been reported [27]. At 10th day, most of the cells were radial in shape and formed a dense area, which was subsequently purified by differential paste-wall method to purify DPCs (Figure 2A). HFSCs gradually migrated from the tissue and grew adherently to the wall at 6th day cultured (Figure 2B). The cells proliferated rapidly and had a flat morphology at 10th days (Figure 2B). After reaching the passaging density, the cells showed a typical cobblestone shape after being purified by the differential paste-wall method (Figure 2B), which was consistent with the reported morphology of HFSCs [28]. Immunofluorescence technique was used to identify HFSCs and DPCs (Figure 2C). DPCs were detected by hepatocyte-specific markers α-SMA and SOX2 and HFSCs by CK14 and CK19, all of which were expressed (Figure 2C). The experimental results show that the DPCs and HFSCs obtained through the separation operation have high purity and can meet the requirements of subsequent research.

### 3.2. Detecting the Transfection Efficiency of FGF22 in DPC

To investigate the effects of *FGF22* on DPCs, we established overexpression and knockout expression systems for *FGF22* and validated their transfection efficiency and functional outcomes (Figure 3). An overexpression plasmid (pcDNA3.1-FGF22) and an empty control group (pcDNA3.1-NC) were transfected and detected efficiency by GFP labeling and observed under fluorescence microscope. Fluorescence microscopy observation of the transfected cell samples showed that over 90% of the cells presented green fluorescence, which directly reflected the successful transfection and high efficiency (Figure 3A). siRNA-FGF22-125, siRNA-FGF22-211, siRNA-FGF22-311, siRNA-FGF22-482 and siRNA-NC were transfected and labeled by FAM-labeled negative control (FAM-NC) and observed under inverted fluorescence microscopy, showed that more than 80% of the cells with green fluorescence, indicating that the transfection was successful (Figure 3B). qRT-PCR results showed that pcDNA3.1-FGF22 in the overexpression group significantly up-regulated the *FGF22* mRNA expression (*p* < 0.01, Figure 3C), and siRNA-FGF22-125 in the Knockout group significantly inhibited the DPCs’ *FGF22* mRNA expression (*p* < 0.01, Figure 3D).

### 3.3. The Effect of FGF22 Overexpression on the Expression Levels of FGF22-Related Receptor Genes and Differentiation-Related Pathway Marker Genes in HFSCs

We explored the effects of *FGF22* overexpression on the receptor genes and the differentiation and pathway marker genes in HFSCs through the co-culture system. *FGFR1*, *FGFR2* mRNA expression was up-regulated in DPCs and HFSCs in the overexpression group (pcDNA3.1-FGF22) compared with that in the empty control group (pcDNA3.1-NC) (*p* < 0.05, Figure 4A,B). In the co-culture system, mRNA expression of Wnt/β-Catenin (β-Catenin, *SOX9*, *c-myc*, *APC*), Shh (*Shh*, *GLI1*, *LGR5*, *Smo*), Notch (*NOTCH1*, *NOTCH3*, *HES1*, *DLL1*) pathway-related marker genes was elevated in HFSCs of the overexpression group, while mRNA expression of BMP pathway-related marker genes (*BMP2*, *BMP4*, *SMAD6*, *ID1*) was decreased (*p* < 0.05, Figure 4C–F). It is suggested that *FGF22* may promote the proliferation and differentiation of HFSCs through overexpression by activating the Wnt/β-Catenin, Shh, Notch pathways and inhibiting the BMP pathway.

### 3.4. Effects of FGF22 Knockout on FGF22-Related Receptor Gene Expression Levels and Differentiation-Related Pathway Marker Genes in HFSCs

We adopted the co-culture method to explore the role of receptor genes related to FGF22-siRNA in cells and their influence on the expression levels of differentiation-related genes and pathway marker genes in HFSCs. The results showed that *FGFR1* and *FGFR2* mRNA expression was down-regulated in DPCs and HFSCs in the FGF22-siRNA group compared with the siRNA-NC group (*p* < 0.05, Figure 5A,B). Similarly, in the FGF22-siRNA group, the mRNA expression levels of marker genes related to differentiation and signaling pathways (including Wnt/β-Catenin, Shh, Notch, etc.) within HFSCs all showed a downward trend (*p* < 0.05, Figure 5C–F). The above research results suggest that *FGF22* knockout may affect the proliferation and differentiation of HFSCs into HFs through the potential mechanism of inhibiting the Wnt/β-Catenin, Shh, Notch pathways and promoting the BMP pathway.

### 3.5. Overexpression of FGF22 in DPCs Promoted the Proliferation of DPCs and HFSCs

The effects of *FGF22* on the proliferation and apoptosis of the co-cultured DPCs and HFSCs model was investigated. The proliferation and apoptosis were detected by EdU, CCK-8, and flow cytometry after co-culture incubated for 24 h. Observation under an inverted fluorescence microscope revealed that overexpression of *FGF22* in DPCs significantly increased the number of EDU-positive cells in both DPCs and HFSCs (*p* < 0.01) (Figure 6A,B). Specifically, the proportion of positive cells increased 2.11-fold in DPCs (*p* < 0.01) and 1.67-fold in HFSCs (*p* < 0.01). CCK-8 assays measured absorbance at 450 nm at 12 h, 24 h, 48 h, and 72 h, with cell proliferation curves plotted. Results demonstrated that compared with the pcDNA3.1-NC group, the proliferation rate of DPCs and HFSCs overexpressing pcDNA3.1-FGF22 was significantly accelerated at 48 h (*p* < 0.01, Figure 6C,D). Flow cytometry analysis indicated that *FGF22* overexpression in DPCs reduced apoptosis rates in both HFSCs and DPCs (*p* < 0.05, Figure 6E,F). Specifically, in DPCs, the proportion of Q2 (late apoptotic cells) in the pcDNA3.1-FGF22 group decreased from 8.60% to 4.64% compared with the pcDNA3.1-NC group, and the proportion of Q4 (early apoptotic cells) decreased from 0.89% to 0.31% (*p* < 0.05, Figure 6E) It was indicated that overexpression of *FGF22* could reduce the early and late apoptosis of DPCs (*p* < 0.05, Figure 6E). In addition, the apoptosis rate of the pcDNA3.1-FGF22 group was significantly lower than that of the pcDNA3.1-NC group (*p* < 0.05), indicating that overexpression of *FGF22* can reduce the apoptosis of DPCs. In HFSCs, the proportion of Q2 in the pcDNA3.1-FGF22 group decreased from 6.12% to 3.79% compared with the pcDNA3.1-NC group. Although the proportion of Q4 slightly increased, the overall proportion showed decreased apoptosis due to a significant reduction in late apoptosis (*p* < 0.05, Figure 6F). Moreover, the apoptosis rate of HFSCs in the pcDNA3.1-FGF22 group was significantly lower than that in the pcDNA3.1-NC group (*p* < 0.05).

In conclusion, overexpression of *FGF22* has a protective effect on DPCs and HFSCs, significantly reducing the apoptosis rate of DPCs and HFSCs, confirming that *FGF22* plays a positive role in maintaining the survival and function of hair follicle cells.

### 3.6. Knockout of FGF22 in DPCs Reduced the Proliferation of DPCs and HFSCs

The effects of *FGF22* on the proliferation and apoptosis of the co-cultured DPCs and HFSCs model was investigated. The proliferation and apoptosis were detected by EdU, CCK-8, and flow cytometry after co-culture incubated for 24 h. Observation under an inverted fluorescence microscope revealed that *FGF22* knockout in DPCs markedly reduced the number of EDU-positive cells in both DPCs and HFSCs (*p* < 0.01) (Figure 6B and Figure 7A). Specifically, the proportion of positive cells decreased by 35% in DPCs (*p* < 0.01) and by 25% in HFSCs (*p* < 0.01). CCK-8 assays measured absorbance at 450 nm at 12 h, 24 h, 48 h, and 72 h, plotting cell proliferation curves. Results demonstrated that compared to the siRNA-NC group, proliferation rates in both DPCs and HFSCs from the siRNA-FGF22 knockout group were markedly slowed at 48 h (*p* < 0.05, Figure 6D and Figure 7C). Flow cytometry analysis revealed that *FGF22* knockout in DPCs increased apoptosis rates in both HFSCs and DPCs (*p* < 0.05; Figure 6F and Figure 7E). Specifically, in DPCs, compared with the siRNA-NC treatment group, the proportion of Q2 (late apoptotic cells) in the siRNA-FGF22 treatment group increased from 4.95% to 11.1%, and the proportion of Q4 (early apoptotic cells) increased from 0.30% to 0.48% (*p* < 0.05, Figure 7C). Moreover, the apoptosis rate in the siRNA-FGF22 group was significantly higher (*p* < 0.05), indicating that knockout of *FGF22* expression could promote the apoptosis of DPCs. In HFSCs, the proportion of Q2 in the siRNA-FGF22 group increased from 4.28% to 7.37% compared with the siRNA-NC group. Although the proportion of Q4 decreased, overall, it showed increased apoptosis due to the increase in late apoptosis (*p* < 0.05, Figure 7D). Moreover, the apoptosis rate of HFSCs in the siRNA-FGF22 group was significantly higher (*p* < 0.05).

In conclusion, *FGF22* is crucial for the survival of DPCs and HFSCs. Knocking out its expression significantly increases the apoptosis rate of both types of cells, suggesting that *FGF22* plays a key role in maintaining the homeostasis of hair follicle cells and anti-apoptosis.

## 4. Discussion

### 4.1. Hair Follicle Cycle Regulatory Networks and Cellular Interactions Mechanisms

HFs are miniature organs undergoing continuous, repetitive growth and cyclical regeneration (Anagen, Catagen, and Telogen) [29], driven by the synergistic action of HFSCs and DPCs and precisely regulated by multiple signaling pathways like Wnt/β-Catenin, Shh, Notch, and BMP [30]. In this study, HFSCs and DPCs were successfully isolated via mechanical separation combined with enzymatic digestion. DPCs, exhibiting pike-shaped, triangular [27], or polygonal morphology with significant agglutination characteristics, highly express α-SMA (a key marker distinguishing them from fibroblasts [28]) and SOX2 (which regulates HF growth rate via the BMP pathway [27]). HFSCs are cobblestone-shaped [31] and specifically express CK14 [32] and CK19 [33]. Immunofluorescence assays confirmed these characteristics, consistent with previous studies [32,34], providing a reliable cellular model for subsequent functional studies.

### 4.2. Functional Validation of FGF22 as a Key Factor in DPCs-HFSCs Interactions

Based on the cell models of HFSCs and DPCs established above, we further investigated the functional role of *FGF22* in DPC-HFSCs interactions. The FGF gene family is widely present in skin tissues and is essential for proliferation, differentiation of DPCs and HFs cyclic growth [35]. *FGF22*, belonging to the *FGF7* subfamily, regulates hair follicle morphogenesis by activating the *FGFR1* and *FGFR2* receptors [25,36]. Relevant reports show that *FGF7* and *FGF10* can stimulate keratinocyte proliferation in normal and injured skin through *FGFR2-IIIb* synergistic effects [37]. In this study, altering *FGF22* expression in DPCs affected *FGFR1/2* mRNA levels in HFSCs, suggesting *FGF22* may mediate paracrine signaling via *FGFR1/2*. Co-culture systems represent a classical model for investigating cellular interactions [38,39,40,41], with prior studies demonstrating that DPCs can dominate HFSC proliferation and differentiation via paracrine mechanisms [42,43]. This study further indicates that *FGF22* overexpression in DPCs promotes HFSC proliferation and inhibits apoptosis in co-culture systems, while *FGF22* knockout produces the opposite effects.

Notably, the findings of this study align with previous research on other members of the FGF family. For example, studies in DPC-HFSCs co-culture systems have demonstrated that *FGF7* secreted by DPCs promotes proliferation of both DPCs and HFSCs via the Wnt signaling pathway and induces HFSC differentiation, indicating that *FGF7* actively drives cell proliferation and differentiation in DPC-HFSC interactions [20]. Furthermore, studies in Merino sheep revealed that *FGF20* knockdown inhibits HFSC proliferation and differentiation in co-culture systems, indicating that *FGF20* is indispensable for maintaining normal HFSC proliferation and differentiation processes in such systems [24]. Studies on different members of the FGF family have revealed similar patterns in regulating the function of HFSCs, verifying the key role of *FGF22* in the interaction between DPC-HFSCs. It has a specific mechanism of action and similar synergy with other members. An in-depth exploration of the FGF22-mediated mechanism is conducive to clarifying the biological principles of hair follicles and providing new theoretical strategies for the treatment of hair follicle diseases and skin tissue engineering.

### 4.3. Molecular Mechanisms of FGF22 in the Multipathway Coordinated Regulation of HFSCs Fate

*FGF22* is vital for regulating HFSC proliferation and apoptosis in DPC-HFSCs interactions. To delve into the underlying mechanisms, we explored the molecular pathways governing HFSC fate. Multiple signaling pathways coordinate to regulate HFSC fate (proliferation, differentiation, quiescence). Wnt/β-Catenin and Notch are classical pathways that promote HFSC proliferation and differentiation and are primary signals during hair regeneration [44]. β-Catenin levels directly influence HFSC differentiation direction [45,46]. Although DPC-secreted *FGF7* reduces β-Catenin expression, it upregulates Wnt pathway downstream genes, promoting HFSC proliferation via enhanced *MYC* expression [20,47]. In our study, *FGF22* overexpression in DPCs significantly up-regulated β-Catenin, c-myc, and APC expression in HFSCs, while interference inhibited the pathway. This suggests *FGF22* may promote HFSC proliferation through the Wnt/β-Catenin pathway.

The Shh signaling pathway is crucial for HFSC regeneration [48]. *Shh* relieves *Smo* inhibition (*Smo* is a GPCR-like protein) to regulate HFSC development [49,50]. Its signaling is similar to Wnt/β-Catenin, affecting dermal/epithelial cell proliferation and epithelial growth and hair papilla formation [51]. We found that *FGF22* overexpression upregulates *Shh*, *GLI1*, *LGR5*, and Smo expression in HFSCs, with the opposite effect upon interference [52]. This implies *FGF22* activates Shh signaling by relieving Smo inhibition, potentially maintaining HFSC stemness and promoting HF regeneration. The Notch signaling pathway maintains the normal cyclic growth of HFs by promoting the differentiation of HFSCs into hair follicle cells and inhibiting the differentiation of epidermal cells [12,53]. *NOTCH1/3*, as the main receptors in HFSCs, with *HES1* as a key downstream gene, and *DLL1* acting as a ligand to activate NOTCH receptors [54]. *FGF22* overexpression upregulated *NOTCH1/3*, *HES1*, and *DLL1* expression in HFSCs, while knockout had the opposite effect. This aligns with Notch signaling’s role in inhibiting epidermal differentiation and promoting HF fate conversion, and explains how *FGF22* knockout suppresses HFSC proliferation by blocking Notch-mediated lateral inhibition. The BMP signaling pathway inhibits proliferation and promotes HF degeneration. *BMP2* (from adipocytes) and *BMP4* (from dermal fibroblasts) inhibit HFSC proliferation [55]. *BMP4* also blocks HF activation from telogen to the early developmental phase [56,57]. SMAD6 represses BMP/Smad1 pathway activity [58], and *ID1* inhibits differentiation but promotes proliferation and migration [59,60]. Our study showed that *FGF22* overexpression in DPCs downregulated *BMP2/4* and *ID1* expression and upregulated *SMAD6* expression in HFSCs, with opposite results upon interference. This indicates *FGF22* may promote HFSC activation and HF regeneration by inhibiting BMP signaling and relieving its suppression on HFSC proliferation.

### 4.4. Limitations and Future Directions of the Research

Although this study has achieved certain results in exploring the biological relationship between *FGF22* and hair, there are still limitations. The selected sheep samples are small and regional, making it difficult to comprehensively reflect biodiversity and ensure the universality of the results. In terms of research design, there is an excessive reliance on in vitro co-culture models, lacking in vivo experiments and organoid verification, making it difficult to comprehensively and deeply evaluate the mechanism of action of *FGF22* and affecting the reliability of clinical translation.

Despite the above-mentioned limitations of the research, the study of *FGF22* in human hair follicle stem cells still holds significant value. On the one hand, it may affect the self-renewal of stem cells by regulating the ECM (extracellular matrix) components of the stem cell niche, which helps us to have a deeper understanding of the biological characteristics of hair follicle stem cells. On the other hand, its competitive binding mechanism may provide new therapeutic targets for androgenetic alopecia. However, this research also faces many challenges. The research results of the animal model need to be verified by the human stem cell system and clinical samples. The functional conservation of *FGF22* varies among species, and multi-omics analysis (including genomics, transcriptomics, proteomics and metabolomics, etc.) is needed to reveal its specific mechanism in humans. Technical challenges such as establishing a 3D culture system that is closer to the in vivo environment and developing precise regulatory strategies still remain to be solved. Subsequent research can be verified and expanded through cross-species comparisons and the use of human samples. At the same time, a complete 3D culture system should be established, precise regulatory strategies should be developed, and the specific mechanism of *FGF22* in humans should be revealed through multi-omics analysis. These subsequent research directions are of great significance for overcoming the limitations of current research, deeply understanding the mechanism of action of *FGF22*, and providing more effective theoretical basis and treatment methods for hair loss treatment.

## 5. Conclusions

This study successfully isolated primary HFSCs and DPCs from fine-wool sheep and established a DPC-HFSC co-culture model, systematically elucidating the pivotal role of *FGF22* secreted by DPCs in regulating HFSC proliferation, differentiation, and apoptosis. Experimental results indicate that *FGF22* overexpression significantly upregulates *FGFR1*/*FGFR2* mRNA expression in HFSCs, promoting their proliferative activity, enhancing cell viability, and inhibiting apoptosis. Conversely, *FGF22* knockout produces opposite effects, leading to reduced HFSC proliferation capacity, increased apoptosis levels, and impaired differentiation. Further investigations revealed that *FGF22* positively regulates HFSC biological behaviors by activating the Wnt/β-Catenin, Sonic Hedgehog (Shh), and Notch signaling pathways, or by inhibiting the BMP signaling pathway. These findings not only elucidate the central role of *FGF22* in DPC-HFSC interactions but also decipher the molecular signaling networks governing HFSC fate determination. This study deepens the understanding of the hair follicle cycle regulation mechanism and provides a theoretical basis for the breeding of fine-wool sheep. Although the revealed molecular mechanism can provide a theoretical basis for future research in hair follicle biology and regenerative medicine, the potential for clinical transformation of its results still requires further verification through in vivo experiments and translational studies.

## Figures and Tables

**Figure 1 biomolecules-15-01560-f001:**
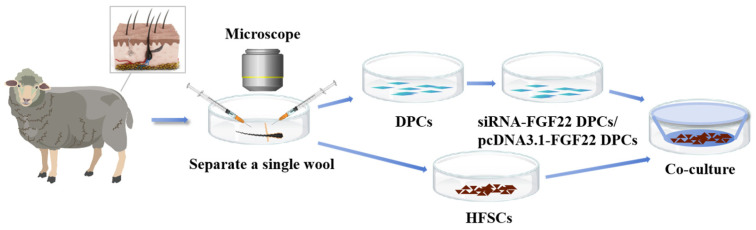
Indirect co-culture model of DPC and HFSC.

**Figure 2 biomolecules-15-01560-f002:**
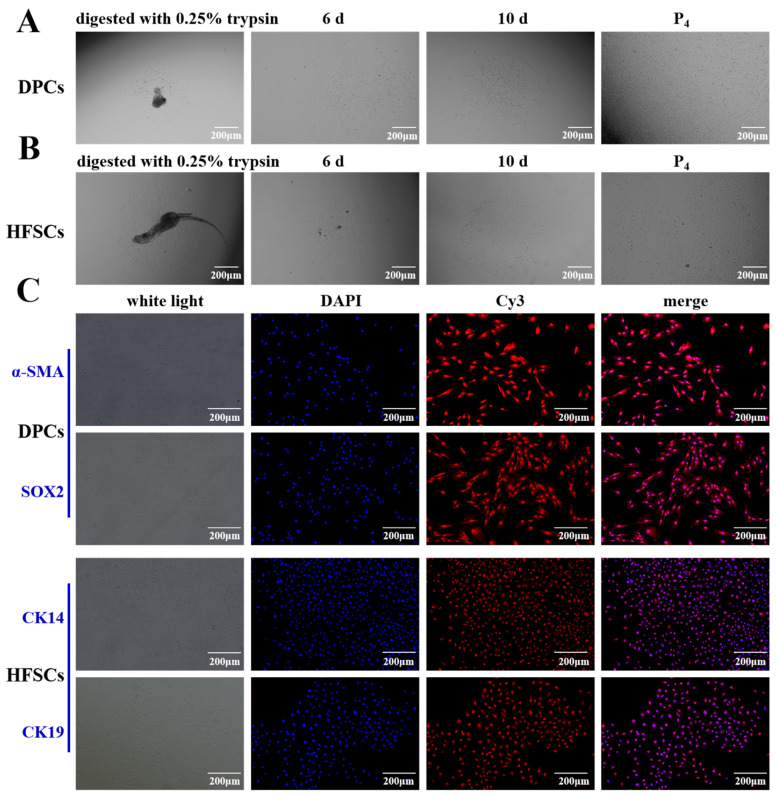
Isolation and characterization of DPCs and HFSCs from fine wool sheep. (**A**) Morphology of primary fine wool sheep HFSCs at 0.25% trypsin digestion, 6th, 10th days and passage 4 (P4). (**B**) After the DPCs of primary fine-wool sheep were digested and purified by trypsin, the morphological characteristics of the cells on the 6th day, the 10th day and the 4th generation were observed (*n* = 3). (**C**) Immunofluorescence staining was carried out on the isolated DPCs and HFSCs (*n* = 3). Antibodies against the specific markers α-SMA and SOX2 of DPCs and CK14 and CK19 of HFSCs were used for labeling. All these markers showed red fluorescence. The bar chart shows the mean ± standard deviation (*n* = 3, biological repetition). Scale bar, 200 μm.

**Figure 3 biomolecules-15-01560-f003:**
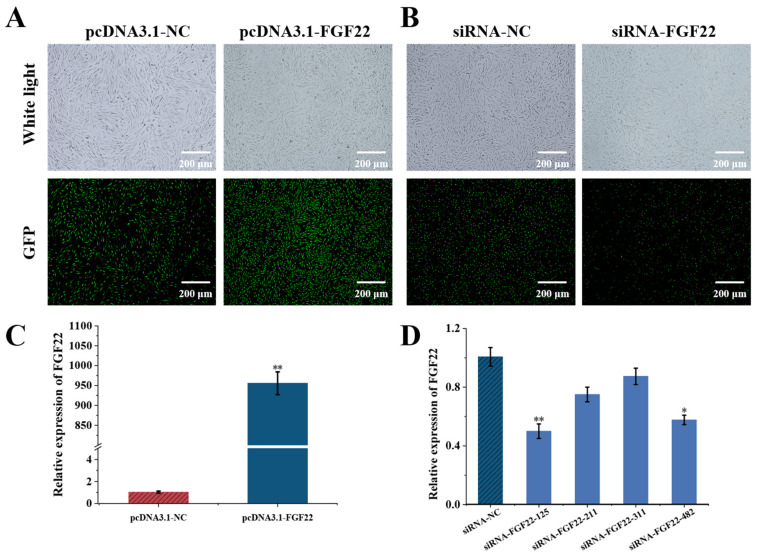
Detection of overexpression and knockout efficiency in DPC. (**A**) Overexpression knockout efficiency of pcDNA3.1-FGF22 in DPC was observed under an inverted fluorescence microscope (*n* = 3). (**B**) Knockout efficiency of siRNA-FGF22 in DPC was observed under inverted fluorescence microscope (*n* = 3). (**C**) qRT-PCR to detect the overexpression efficiency of pcDNA3.1-FGF22 (*n* = 3). (**D**) qRT-PCR to detect the transfection efficiency of siRNA-FGF22 (*n* = 3). The bar chart shows the mean ± standard deviation (*n* = 3, biological repetition). ** indicates *p* < 0.01, * indicates *p* < 0.05. Scale bar, 200 μm.

**Figure 4 biomolecules-15-01560-f004:**
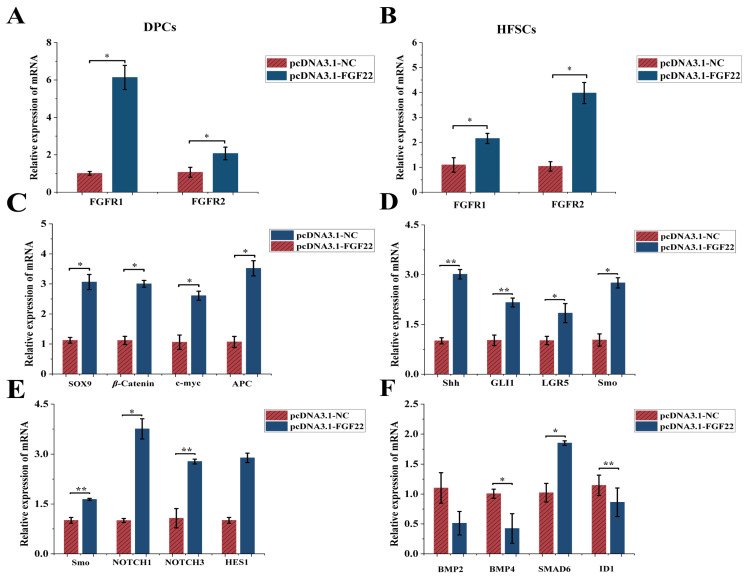
The mRNA expression levels of related receptor genes and pathway marker genes when FGF22 was overexpressed in the co-culture system were detected by qRT-PCR technology. (**A**) The relative expression levels of *FGFR1* and *FGFR2* in DPCs were detected after overexpression of *FGF22* transfection (*n* = 3). (**B**) The relative expression levels of *FGFR1* and *FGFR2* in HFSCs were detected after overexpression of *FGF22* transfection (*n* = 3). (**C**) After transfection of DPCs with overexpressed *FGF22*, the effect of HFSCs differentiation-related gene expression in the Wnt/β-Catenin pathway was determined in a co-culture system (*n* = 3). (**D**) After transfection of DPCs with overexpressed *FGF22*, the effect of HFSCs differentiation-related gene expression in the Shh pathway was determined in a co-culture system (*n* = 3). (**E**) After transfection of DPCs with overexpressed *FGF22*, the effect of HFSCs differentiation-related gene expression in the Notch pathway was determined in a co-culture system (*n* = 3). (**F**) After transfection of DPCs with overexpressed *FGF22*, the effect of HFSCs differentiation-related gene expression in the BMP pathway was determined in a co-culture system (*n* = 3). The bar chart shows the mean ± standard deviation (*n* = 3, biological repetition). ** indicates *p* < 0.01, * indicates *p* < 0.05.

**Figure 5 biomolecules-15-01560-f005:**
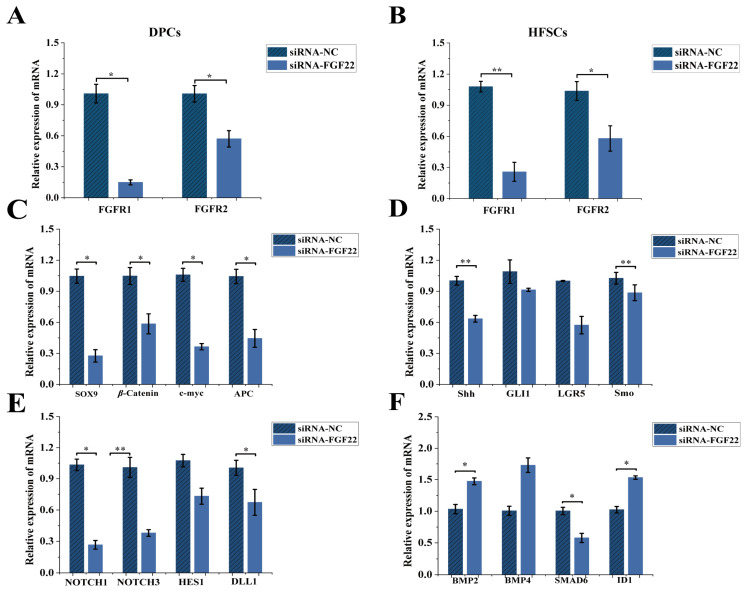
The mRNA expression levels of related receptor genes and pathway marker genes when *FGF22* was overexpressed in the co-culture system were detected by qRT-PCR technology. (**A**) The relative expression levels of *FGFR1* and *FGFR2* in HFSCs were detected after *FGF22* knockout transfection (*n* = 3). (**B**) The relative expression levels of *FGFR1* and *FGFR2* in DPCs were detected after *FGF22* knockout transfection (*n* = 3). (**C**) After transfection of DPCs with overexpressed *FGF22*, the effect of HFSCs differentiation-related gene expression in the Wnt/β-Catenin pathway was determined in a co-culture system (*n* = 3). (**D**) After transfection of DPCs with overexpressed *FGF22*, the effect of HFSCs differentiation-related gene expression in the Shh pathway was determined in a co-culture system (*n* = 3). (**E**) After transfection of DPCs with overexpressed *FGF22*, the effect of HFSCs differentiation-related gene expression in the Notch pathway was determined in a co-culture system (*n* = 3). (**F**) After transfection of DPCs with overexpressed *FGF22*, the effect of HFSCs differentiation-related gene expression in the BMP pathway was determined in a co-culture system (*n* = 3). The bar chart shows the mean ± standard deviation (*n* = 3, biological repetition). ** indicates *p* < 0.01, * indicates *p* < 0.05.

**Figure 6 biomolecules-15-01560-f006:**
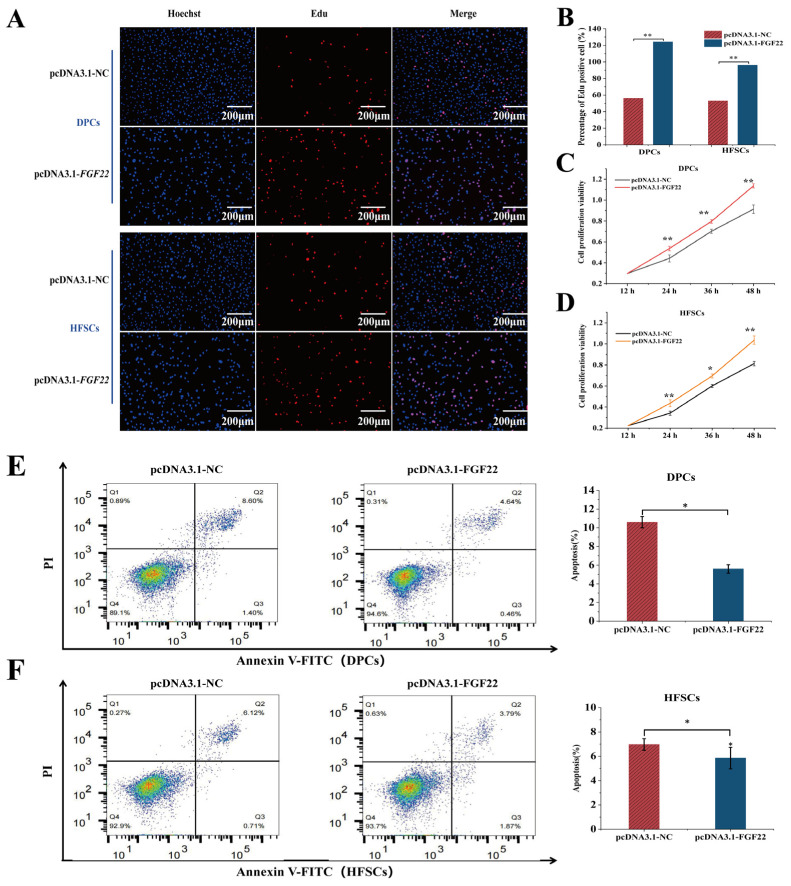
Detection of pcDNA3.1-FGF22 on the number of DPCs-positive cells, cell viability and apoptosis under co-culture conditions. (**A**) The EdU staining effects of HFSCs and DPCs were observed and analyzed under an inverted fluorescence microscope (*n* = 3). (**B**) Statistical analysis of the number of EdU positive cells in DPCs and HFSCs (*n* = 3). (**C**) The cell viability of DPCs was quantitatively detected and analyzed using the CCK-8 kit (*n* = 3). (**D**) The cell viability of HFSCs was quantitatively detected and analyzed using the CCK-8 kit (*n* = 3). (**E**) The apoptosis rate of DPCs was detected by flow cytometry (*n* = 3). (**F**) The apoptosis rate of HFSCs was detected by flow cytometry (*n* = 3). The bar chart shows the mean ± standard deviation (*n* = 3, biological repetition). ** indicates *p* < 0.01, * indicates *p* < 0.05. Scale bar, 200 μm.

**Figure 7 biomolecules-15-01560-f007:**
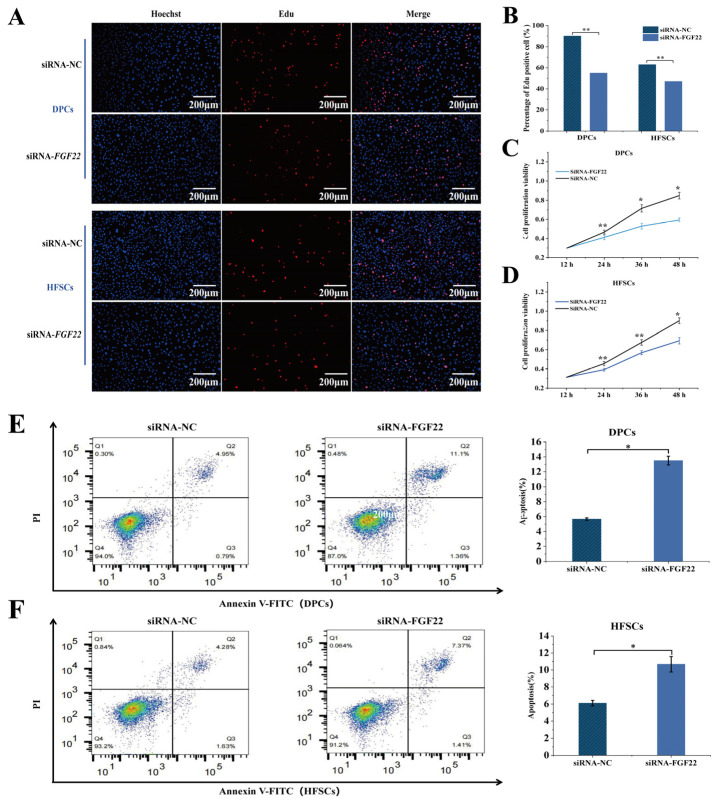
Under the HFSC-DPC co-culture system, the effects of the number of DPC-positive cells, cell viability and apoptosis were detected. (**A**) The EdU staining effects of HFSCs and DPCs were observed and analyzed under an inverted fluorescence microscope (*n* = 3). (**B**) Statistical analysis of the number of EdU positive cells in DPCs and HFSCs (*n* = 3). (**C**) The cell viability of DPCs was quantitatively detected and analyzed using the CCK-8 kit (*n* = 3). (**D**) The cell viability of HFSCs was quantitatively detected and analyzed using the CCK-8 kit (*n* = 3). (**E**) The apoptosis rate of DPCs was detected by flow cytometry (*n* = 3). (**F**) The apoptosis rate of HFSCs was detected by flow cytometry (*n* = 3). The bar chart shows the mean ± standard deviation (*n* = 3, biological repetition). ** indicates *p* < 0.01, * indicates *p* < 0.05. Scale bar, 200 μm.

**Table 1 biomolecules-15-01560-t001:** Preparation Reagents for Cell Culture Medium.

	DMEM/F12 Medium	10% FBS	5 µg/mL Insulin	10 µg/mL Epidermal Growth Factor	0.5 µg/mL Hydrocortisone
DPCs	√	√	√	√	
HFSCs	√	√	√	√	√

“√” indicates that the corresponding reagents have been added to the DPCs or HFSCs culture system.

**Table 2 biomolecules-15-01560-t002:** RNAoligonucleotide synthetic sequence.

Gene Name	Sequence (5′-3′)
siRNA-FGF22-125	F: CCUAUGGCUGGGCCUGGUGTT
	R: CACCAGGCCCAGCCAUAGGTT
siRNA-FGF22-211	F: ACCCGCACCUGGAGGGCGATT
	R: UCGCCCUCCAGGUGCGGGUTT
siRNA-FGF22-311	F: GCGCGACAACCCUGACAGCTT
	R: GCUGUCAGGGUUGUCGCGCTT
siRNA-FGF22-482	F: CUACAAUACCUACGCGUCATT
	R: UGACGCGUAGGUAUUGUAGTT
siRNA-NC	F: UUCUCCGAACGUGUCACGUTT
	R: ACGUGACACGUUCGGAGAATT

**Table 3 biomolecules-15-01560-t003:** qRT-PCR primers.

Gene Name	Forward Primer Sequence	Reverse Primer Sequence
FGF22	GAGCCCGTGTCCAGTTACTC	ACCACCCCTCCAACTCAGT
FGFR1	ACTCTACCCCAGCCCTAAGG	ACCATCCATTCACACGACCC
FGFR2	AGAGTGATGTCTGGTCCTTCG	GAAGAGCAAGAACTCCTGGTG
β-Catenin	ACACAGTTCGATGCTGCTCA	GATTGCACGTGTGGCAAGTT
SOX9	CGAAACTGGACTGGAAACCT	GTTCTCTCTGCCTGTTTGGA
c-myc	CGGAAGAGGCGAGAACAGTT	ATCCAGCCAAGGTTGTGAGG
APC	TCCTCCCGACTCAGTGTTCT	GTCCCCGTCGCTATACTTGG
Shh	TGGTCTCCTCGCTGTTGATG	CTCGTATCTTCCACTGGCCC
GLI1	CCTCGTGGCCTTCATCAACT	GGTGGTTCATCTGGGGTGAG
LGR5	TTCACTTTCGGCAGCTTTGC	CCAGGGTGAGCAGGAAAACT
Smo	CTCCCTTGGTTCGGACTGAC	GTAGCTGTGCATGTCCTGGT
NOTCH1	ACAACGCCTACCTCTGCTTC	ACACATGCTCCCTGTGTAGC
NOTCH3	CTTGGGTCCTGTGGTGAGTC	AGCAGGAGGAGTGAGAGAGG
HES1	GACCCAGATCAACGCCATGA	GAACACCTTAGCCGCCTCTC
DLL1	GCACGGACCTCAAGTACTCC	ACTTTCTCCCCTCTCTCCCC
BMP2	CACACCCTACCCGAGATTGG	CTGAGTCCCCAGTAATCCGC
BMP4	CTTCAGTCTGGGGAGGAGGA	AGAGTTTTCGCTGGTCCCTG
SMAD6	AACCCCTACCATTTCAGCCG	GGAGTTGGTGGCCTCTGTTT
ID1	AAGGTGAGCAGGGTGGAGAT	TCGCCATTGAGAGTGCTGAG
β-actin	CAGTCGGTTGGATGGAGCAT	AGGCAGGGACTTCCTGTAAC

## Data Availability

All data is presented in the article.

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
