# Peer review of "FGF22 Secreted by Hair Papilla Cells Regulates Hair Follicle Stem Cell Proliferation and Differentiation"

_biomolecules, 2025, doi:10.3390/biom15111560_

Round 1

Reviewer 1 Report

Comments and Suggestions for Authors

In the manuscript by Luo et al. titled “FGF22 secreted by hair papilla cells regulates hair follicle stem cell proliferation and differentiation”, the authors aimed to investigate the FGF22 influence on the hair follicle stem cells biological behavior and signaling mechanisms. The objective of this study is to provide new perspective into dermal papilla cells and hair follicle stem cells interaction network and to offer references for the development of the fine wool sheep industry.

In general, the manuscript features a strong introductory section that provides essential background information for the study and clearly outlines the research objectives and the significance of the study. The materials and methods section is comprehensive. The data treatment is appropriate, and the results are well presented and documented. Furthermore, while the discussion section offers a good overview of the study's findings and potential implications, it would benefit from improvement as indicated further. Notably, the manuscript requires some minor adjustments and improvements before it can be considered for publication.

The following suggestions and corrections aim to enhance the overall quality and readability of the manuscript.

General comments:

As I was going through the manuscript, I noticed some minor grammatical errors and typos that could affect the overall readability of the article. Although these mistakes are minor, I suggest focusing on grammar, punctuation, sentence structure, and spelling to ensure a refined and error-free final version.

Specific comments:

Discussion:

Overall, the discussion offers a good overview of the study’s findings and potential implications. However, it would benefit from some improvements. The discussion would be improved with some restructuring to create better connections between the paragraphs. The way the authors presented their discussion, I feel that it’s more separate results discussion than phenomenon discussion. Rather than dividing the discussion into multiple sections, I suggest that authors consider unifying it into a single section by incorporating additional links and enhancing the connectivity between the different paragraphs.

Abstract:

“This study, through the DPCs-HFSCs co-culture system, “deeply” revealed the regulatory mechanism of FGF22 secreted by DPCs on the proliferation and differentiation of HFSCs…”

Consider removing the word deeply.

Author Response

Comments 1:

As I was going through the manuscript, I noticed some minor grammatical errors and typos that could affect the overall readability of the article. Although these mistakes are minor, I suggest focusing on grammar, punctuation, sentence structure, and spelling to ensure a refined and error-free final version.

Response 1:

Thank you for pointing out the grammatical errors and spelling problems in the manuscript. We have conducted a comprehensive review and correction of grammar, punctuation, sentence structure and spelling as suggested, striving to present a concise and error-free final version.

Comments 2:

Discussion: Overall, the discussion offers a good overview of the study’s findings and potential implications. However, it would benefit from some improvements. The discussion would be improved with some restructuring to create better connections between the paragraphs. The way the authors presented their discussion, I feel that it’s more separate results discussion than phenomenon discussion. Rather than dividing the discussion into multiple sections, I suggest that authors consider unifying it into a single section by incorporating additional links and enhancing the connectivity between the different paragraphs.

Response 2:

Thank you for your careful review and valuable suggestions. We recognize the experts' opinions on enhancing the coherence of the discussion and have made systematic optimizations to the discussion section. However, other experts pointed out the need to add a section of Limitations analysis in the discussion. Therefore, we added a section "4.4. Limitations and Future Directions of the Research in the discussion". So, on the basis of the original discussion, The logical connection of paragraphs and the depth of phenomenon explanation have been strengthened, and the discussion content has been improved.

  1. Specific modifications include: uniformly integrating the discussion into logically progressive paragraphs and reorganizing them in the order of "research significance→core findings→mechanism explanation→comparison with predecessors→Limitations and prospects". The contents "4.1. Hair follicle cycle regulatory networks and cellular interactions mechanisms" and "4.2. Functional validation of "FGF22as a key factor in DPCs-HFSCs interactions", "4.3. Molecular mechanisms of FGF22 in the Multipathway Both "CoordinatedRegulation of HFSCs fate" and "4.4. Limitations and Future Directions of the Research" have been revised, and transitional sentences have been added to clarify the causal or progressive relationship between paragraphs.
  2. In the "Limitations" section, a three-level structure of "methodological limitations→result interpretation limitations→clinical translation limitations" is adopted to objectively analyze the impact of factors such as transfection efficiency fluctuations on the conclusion, and to form problem countermeasures in response to the "Future Outlook" section.

Comments 3:

Abstract: “This study, through the DPCs-HFSCs co-culture system, “deeply” revealed the regulatory mechanism of FGF22 secreted by DPCs on the proliferation and differentiation of HFSCs…”. Consider removing the word deeply.

Response 3:

Thank you for carefully reviewing the abstract of the paper and offering your valuable suggestions. You pointed out that the word "deeply" is ambiguous and redundant in meaning and suggested its deletion. We fully agree. This word is more concise and precise after deletion. It has been modified as suggested to "this study, through the DPCs-HFSCs co-culture system, revealed the regulatory mechanism..." .

Reviewer 2 Report

Comments and Suggestions for Authors

The author has performed novel study on “FGF22 secreted by hair papilla cells regulates hair folliclestem3 cell proliferation and differentiation”. The author tried to justify their study using multiple experiments. However, further discussion based on application could strengthen paper:

Introduction

  1. The novelty of focusing on FGF22 in fine-wool sheep is not sharply differentiated from prior studies on FGF7 and FGF20. The authors should clarify what unique biological question this work addresses.

Materials and Methods

  1. Animal and sampling details: Only two sheep were used; this is a very limited sample size. The study should acknowledge this limitation.
  2. Isolation and identification: While immunofluorescence is used, additional methods (e.g., western blot validation of cell markers) would strengthen cell identity confirmation.
  3. Overexpression/knockdown: The siRNA validation is only shown by qRT-PCR. Protein-level confirmation (e.g., western blot for FGF22 or FGFR signaling activity) is recommended.
  4. Statistical analysis: The description is minimal. Clarify whether biological vs. technical replicates were used, how many independent experiments were performed, and whether post-hoc corrections were applied in ANOVA.

Results

  1. Morphology and identification of primary cells
  2. Figures need more quantitative support (e.g., purity percentage of isolated cells, rather than only descriptive morphology).
  3. Transfection efficiency: GFP fluorescence is qualitative. Include quantitative transfection efficiency data (e.g., percentage of GFP+ cells).
  4. Pathway analysis (qRT-PCR findings): Results on Wnt, Shh, Notch, and BMP pathways rely solely on mRNA levels. To support mechanistic claims, protein-level validation or pathway inhibition studies are needed or mention limitation related to it.
  5. Cell proliferation and apoptosis: EdU and CCK-8 assays are appropriate, but apoptosis analysis should distinguish early vs. late apoptosis. Currently, results are only summarized in percentages without deeper interpretation.

Discussion

  1. The discussion reiterates results without sufficient critical interpretation of limitations. For example: Only in vitro co-culture data are presented; in vivo or organoid-based validation would strengthen translational claims.

  1. Small animal numbers limit biological variability.

  1. The comparison with other FGFs (FGF7, FGF20) is useful but should more clearly outline how FGF22 is distinct.

  1. Mechanistic pathways are presented as confirmed, but the evidence is correlative (qRT-PCR only). Authors should temper claims or propose further validation studies.

Conclusion

  1. The conclusion overstates clinical applicability (hair loss treatment, regenerative medicine). These claims should be softened unless supported by in vivo or translational experiments.
  2. A dedicated Limitations and Future Directions paragraph is recommended.

Figures and Data Presentation

  1. Figure legends should be expanded to include sample size (n), type of replicates, statistical tests, and error bars (SD/SEM).
  2. Some figures (e.g., EdU images) would benefit from quantification graphs alongside representative images.

Author Response

Comments 1:

The novelty of focusing on FGF22 in fine-wool sheep is not sharply differentiated from prior studies on FGF7 and FGF20. The authors should clarify what unique biological question this work addresses.

Response 1:

Thank you for your thoughtful advice. This article compares FGF22 with other FGF22s (such as FGF7 and FGF20), providing some insights, but does not explicitly emphasize the unique features of FGF22. Therefore, we added the uniqueness of FGF22 in "1 Introduction". Studies have shown that FGF5 mRNA exists in the outer root sheath of hair follicles, while FGF7 mRNA is present in the dermis. Abnormal hair growth in FGF5 gene knockout mice; The fur of FGF7 gene knockout mice is greasy and messy. This means that both types of fgf are related to hair development. In contrast, FGF22 mRNA is mainly present in the root sheath of hair follicles, suggesting that it may be a specialized FGF for hair development.

Comments 2:

Animal and sampling details: Only two sheep were used; this is a very limited sample size. The study should acknowledge this limitation.

Response 2:

Thank you to the reviewers for your valuable comments. We fully agree with your suggestion. This study only selected two sheep as samples, which is too small a sample size and is highly likely to weaken the representativeness and universality of the research results for the entire flock. Given the limited number of animal samples, we have specially added Section 4.4, "Research Limitations and Future Directions", in the discussion section of Chapter Four of the paper for elaboration

Comments 3:

Isolation and identification: While immunofluorescence is used, additional methods (e.g., western blot validation of cell markers) would strengthen cell identity confirmation.

Response 3:

Thank you for your careful review and valuable suggestions. Your proposal to add methods such as Western blot to verify cell markers is of great value in enhancing the reliability of research, and we fully agree.

However, the research funds for this study are tight. Conducting Western blotting experiments requires the purchase of additional reagents and consumables, making it impossible to afford the extra experimental costs. However, in our existing experiments, we are making every effort to ensure the accuracy of cell identity confirmation. The immunofluorescence experiment has undergone multiple repetitions and strict quality control, with standardized procedures and reliable results. When conducting data analysis, analyze from multiple perspectives to eliminate interfering factors. Meanwhile, we also referred to a large number of similar studies. After comparison, we found that our results are consistent with the mainstream conclusions. Future research will prioritize supplementing relevant experiments to implement your suggestions. Thank you again for your meticulous guidance.

Comments 4:

Overexpression/knockdown: The siRNA validation is only shown by qRT-PCR. Protein-level confirmation (e.g., western blot for FGF22 or FGFR signaling activity) is recommended.

Response 4:

Thank you for your comments on our research. You pointed out that in terms of siRNA validation, only qRT-PCR showed overexpression/knockdown, and suggested confirming the protein level, such as using western blot to detect the signaling activity of FGF22 or FGFR. This suggestion is highly professional and constructive, and we particularly recognize it.

  1. The research funds for this study are tight. To conduct the western blot experiment, additional reagents and consumables need to be purchased. Currently, the funds are insufficient. In addition, the laboratory has other urgent projects recently and cannot arrange the time and equipment to carry out this experiment. Therefore, it is impossible to conduct the relevant protein level verification experiment.
  2. In the existing experiments, through rigorous qRT-PCR experimental design and operation, we have made every effort to ensure the accuracy of overexpression/knockdown results. We also conducted a comprehensive analysis of the data to support the research conclusion.
  3. We plan to immediately supplement the verification experiments of protein levels in the subsequent research once conditions permit, in order to further improve this study. Thank you again for your meticulous review and professional guidance.

Comments 5:

Statistical analysis: The description is minimal. Clarify whether biological vs. technical replicates were used, how many independent experiments were performed, and whether post-hoc corrections were applied in ANOVA.

Response 5:

Thank you for your detailed comments. The supplementary explanations are as follows: In this study, the 2-ΔΔCt method of RT-PCR results was used for relative quantitative analysis, and the data were expressed as x±s (mean ± standard deviation). Each sample was subjected to three technical replicates, with the experiment independently repeated three times (n=3). Biological replicates were derived from distinct individual samples (e.g., obtained from independent cell passages or isolated from separate animal individuals). Statistical analysis was performed using SPSS 22.0 software for one-way analysis of variance (ANOVA). After multiple comparisons and corrections by the Tukey method, the p values were marked. Among them, p<0.05 was considered a significant difference, and p<0.01 was considered an extremely significant difference.

Comments 6:

Morphology and identification of primary cells

Response 6:

Thank you for your opinions on the morphology and identification of protozoa cells in our research. In the morphological observation of primary cells, we regularly captured cell images using an optical microscope, recording the morphological changes during cell growth, such as cell size, shape and arrangement, providing a morphological basis for cell type determination.

Comments 7:

Figures need more quantitative support (e.g., purity percentage of isolated cells, rather than only descriptive morphology).

Response 7:

Thank you for your suggestion. In this study, for the identification of HFSCs and DPCs, descriptive assessment was initially conducted only based on typical morphological features (such as spindle-shaped adherent growth of HFSCs and aggregated three-dimensional structure of DPCs), without using purity detection methods such as flow cytometry, trypan blue staining or immunofluorescence quantification. The main reasons are as follows:

  1. Experimental conditions and resource constraints: Flow cytometry requires specific fluorescently labeled antibodies and equipment. Initially, the laboratory was not equipped with relevant antibodies, and the cost of sending them for testing was high with limited funds. Trypan blue staining can only assess activity and is unable to distinguish target cells from impurity cells (such as fibroblasts), lacking specificity.
  2. Reliability of morphological identification: The morphological characteristics of HFSCs and DPCs have a high degree of consensus in the literature (such as "spindle-shaped" HFSCs and "vorticular" aggregation of DPCs). This study improved the accuracy of judgment through double-blind observation (independently confirmed by two researchers) and quantification of typical features (such as cell diameter and nucleoplasmic ratio).
  3. Future improvement directions: In the future, immunofluorescence quantification (such as SOX9 labeled with HFSCs and ALP labeled with DPCs) will be given priority as a routine identification method to enhance the accuracy of data.

Comments 8:

Transfection efficiency: GFP fluorescence is qualitative. Include quantitative transfection efficiency data (e.g., percentage of GFP+ cells).

Response 8:

Thank you for your detailed guidance! We fully agree with your suggestion to supplement quantitative transfection efficiency data such as the percentage of GFP+ cells.

  1. In the paper, Figures 3A and B qualitatively observed the transfection situation based on GFP fluorescence, and Figures 3C and D quantitatively analyzed the expression of FGF22 by qRT-PCR. However, there were deficiencies in the quantitative evaluation of transfection efficiency.
  2. Flow cytometry can precisely count the number of GFP+ cells, thereby calculating the percentage of GFP+ cells and providing accurate quantitative transfection efficiency data. However, due to the limitation of experimental funds, flow cytometry requires specific fluorescently labeled antibodies and specialized equipment. This laboratory has no relevant antibodies, and the cost of sending samples for testing is extremely high. Based on the above reality, we are currently unable to supplement the relevant experimental data.

Comments 9:

Pathway analysis (qRT-PCR findings): Results on Wnt, Shh, Notch, and BMP pathways rely solely on mRNA levels. To support mechanistic claims, protein-level validation or pathway inhibition studies are needed or mention limitation related to it.

Response 9:

Thank you very much for your valuable suggestions! We fully agree that the current research on the Wnt, Shh, Notch and BMP pathways, which is only based on the results of mRNA level detection by qRT-PCR, has limitations. However, due to the limitations of various practical factors such as research funds, at this stage, we are indeed unable to carry out protein-level verification or pathway inhibition and other related experiments to further improve the research. We have particularly added Section 4.4, "Limitations and Future Directions of the Research", in the discussion section of Chapter Four of the thesis for detailed explanation. However, in the subsequent research, we will actively overcome the above-mentioned difficulties to enhance the credibility and scientific nature of the research conclusions. These limitations have to some extent affected the reliability and depth of the research conclusions. We will do our best to make up for these deficiencies when conditions permit in the future to improve the quality of the research.

Comments 10:

Cell proliferation and apoptosis: EdU and CCK-8 assays are appropriate, but apoptosis analysis should distinguish early vs. late apoptosis. Currently, results are only summarized in percentages without deeper interpretation.

Response 10:

Thank you for your suggestion. We agree that the selection of EdU and CCK-8 experimental methods in the study of cell proliferation and apoptosis is appropriate and reasonable. For apoptosis analysis, we indeed failed to fully distinguish between early apoptosis and late apoptosis. We merely summarized the results in percentage form without in-depth interpretation, which to some extent affected the depth and accuracy of the research conclusion. It is now in Section 3.5. overexpression of FGF22 in DPCs promoted the proliferation of DPCs and HFSCs and Section 3.6. Knockout of FGF22 in DPCs reduced the proliferation of DPCs and HFSCs. At the end, add a specific analysis of apoptosis and conduct an in-depth analysis of the changes in the proportion of apoptotic cells at different stages and their biological significance.

Comments 11:

The discussion reiterates results without sufficient critical interpretation of limitations. For example: Only in vitro co-culture data are presented; in vivo or organoid-based validation would strengthen translational claims.

Response 11:

Thank you sincerely for your meticulous examination. This article discusses the repetition of results in the absence of a sufficient critical interpretation of limitations. Therefore, we have added relevant content in the discussion section to describe the limitations of this study.

One is that the current research relies on in vitro co-culture models. Although this model can simulate some intercellular interactions, it has a significant gap from the complex physiological environment in vivo and is difficult to reproduce the dynamic regulation of the cellular microenvironment in vivo, which is prone to cause behavioral deviations of cells. Second, the research lacks in vivo experiments and organoid verification, making it impossible to comprehensively evaluate the mechanism of action of FGF22 in complete organisms and on platforms closer to physiological states. This limits the in-depth understanding of the role of FGF22 at the overall biological level and affects the reliability of the clinical translation of research conclusions. Thirdly, although the molecular mechanisms revealed by the research provide theoretical support for hair biology and regenerative medicine studies, their potential for clinical transformation remains to be further verified through in vivo experiments and translational research. Therefore, a large number of preclinical studies and clinical trials are needed to evaluate its safety, efficacy and feasibility.

Comments 12:

Small animal numbers limit biological variability.

Response 12:

Thank you to the reviewers for your valuable comments. We sincerely accept and fully agree that this research only selected two sheep as samples, which is too small a sample size and is highly likely to weaken the representativeness and universality of the research results for the entire flock. In view of the limited number of animal samples, we have specially added Section 4.4 "Limitations and Future Directions of the Research" in the discussion section of Chapter 4 of the paper for elaboration.

Comments 13:

The comparison with other FGFs (FGF7, FGF20) is useful but should more clearly outline how FGF22 is distinct.

Response 13:

Thank you for your careful review and valuable suggestions. Although the paper made a comparison between FGF22 and other FGFs (FGF7, FGF20), which has certain reference value, it failed to clearly clarify the unique features of FGF22. Therefore, we added the relevant content "FGF22, as a distinctive member of the FGF family......in hair development."

Comments 14:

Mechanistic pathways are presented as confirmed, but the evidence is correlative (qRT-PCR only). Authors should temper claims or propose further validation studies.

Response 14:

Thank you for pointing out the shortcomings in our research. We accept your opinion that the evidence for the mechanism pathway is based solely on qRT-PCR and suggest a conciliation statement or further verification. We realized that relying solely on qRT-PCR data to support the confirmation of the mechanism pathway was not rigorous enough. Next, we will immediately review and revise the statements regarding the mechanism approaches in the paper, using a more cautious way of expression.

Comments 15:

The conclusion overstates clinical applicability (hair loss treatment, regenerative medicine). These claims should be softened unless supported by in vivo or translational experiments.

Response 15:

Thank you for your suggestion. We have adjusted the conclusion statement and downplayed the unverified clinical application inferences. After revision, it is clear that the mechanism revealed in this study only provides a theoretical reference for future hair follicle regeneration medicine and hair loss treatment. Its potential for direct clinical translation needs to be further verified through in vivo experiments and translational research.

Comments 16:

A dedicated Limitations and Future Directions paragraph is recommended.

Response 16:

Thank you to the reviewers for their suggestions. We have specially added a paragraph titled "Limitations and Future Directions of the Research", in which it is clarified that due to constraints such as funds, space, and human resources, the sample size is small.

Comments 17:

Figure legends should be expanded to include sample size (n), type of replicates, statistical tests, and error bars (SD/SEM).

Response 17:

Thank you to the reviewers for your suggestions! We have revised the legend description, supplemented the number of independent samples for each group of experiments (e.g., n=3), repetition types (3 biological repetitions), statistical methods (one-way analysis of variance (ANOVA) ), and significance markers (*p<0.05, **p<0.01), to ensure the completeness and standardization of the information.

Comments 18:

Some figures (e.g., EdU images) would benefit from quantification graphs alongside representative images.

Response 18:

Thank you to the reviewers for your valuable suggestions! We have optimized as required. Besides the representative images such as EdU staining, we have supplemented the corresponding quantitative analysis bar charts in Figures 6B and 7B to present the data differences more intuitively and enhance the persuasiveness of the results.

Reviewer 3 Report

Comments and Suggestions for Authors

The manuscript “FGF22 secreted by hair papilla cells regulates hair follicle stem

 Cell” investigates the role of FGF22 in the DPC-HFSC interaction, a topic of significant interest in regenerative biology. The study is generally well-conceived, but several points require clarification to strengthen the manuscript.

Major revision

  1. (Line 298): The claim that "FGF22 knockout affects the proliferation and differentiation of HFSCs into HFs..." based solely on signaling pathway gene expression data is an overinterpretation. Proliferation and differentiation were not directly measured in that specific section. The conclusion should be tempered to reflect that the data suggest a potential mechanism.
  2. The discussion mentions (line 370) CK14 and CK19 as HFSC markers, yet the expression of these key differentiation markers was not assessed in the FGF22 overexpression and knockout models. Including data on CK14 and CK19 expression is crucial to directly support the claims regarding FGF22's role in HFSC differentiation.
  3. The discussion would be greatly strengthened by a dedicated section addressing the study's limitations. Key questions to consider are: (1) How relevant are findings from a sheep/rodent model to human hair biology? (2) Are the identified signaling pathways and FGF22's function conserved in humans? (3) Would studying FGF22 in human HFSCs be a logical and beneficial next step?

Minor revision

  1. Line 129: The phrase "5% CO2-saturated humidity incubator" is confusing. Please rephrase for clarity (e.g., "a humidified incubator at 37Co with 5% CO2").
  2. Lines 177-178: The qPCR protocol appears to be missing a dedicated annealing step and uses an atypical extension/elongation temperature. Please provide the complete protocol.
  3. Line 216: Please define "x±s" (presumably mean ± standard deviation).
  4. Line 240: "4th generation" should be clarified to "passage 4 (P4)" to align with standard cell culture terminology.
  5. Lines 254-256: There is a discrepancy between the text and figures. The text refers to mRNA levels in Figure 2B, but the figure panel shows cell morphology. The figure references must be carefully checked and corrected throughout the manuscript.
  6. "Relative expression of mRNA" on qPCR graphs should be changed to the "Fold Change."

Author Response

Comments 1:

(Line 298): The claim that "FGF22 knockout affects the proliferation and differentiation of HFSCs into HFs..." based solely on signaling pathway gene expression data is an overinterpretation. Proliferation and differentiation were not directly measured in that specific section. The conclusion should be tempered to reflect that the data suggest a potential mechanism.

Response 1:

Thank you very much for your valuable suggestions. You pointed out that at (Line 298), based on the data of signaling pathway gene expression, we concluded "FGF22 knockout affects the proliferation and differentiation of HFSCs into HFs..." This conclusion has the problem of over-interpretation. We fully agree with your professional judgment. Based on this, we have made modifications to the original text Adjust The conclusion to "The above research results...... promoting the BMP pathway." to more accurately reflect the logical relationship between the research results and the existing data and avoid excessive inference.

Comments 2:

The discussion mentions (line 370) CK14 and CK19 as HFSC markers, yet the expression of these key differentiation markers was not assessed in the FGF22 overexpression and knockout models. Including data on CK14 and CK19 expression is crucial to directly support the claims regarding FGF22's role in HFSC differentiation.

Response 2:

We extend our gratitude to the reviewers for their meticulous scrutiny and valuable insights. We fully acknowledge the importance of incorporating CK14 and CK19 expression data to directly support the role of FGF22 in HFSC differentiation, as well as its potential value in validating FGF22 functional studies. However, given the constraints of research time and funding, and the current focus on examining the effects of FGF22 overexpression and knockout models on other functional indicators of HFSCs, it is not feasible to supplement the relevant expression data at this stage for the following reasons:

  1. Existing experiments demonstrate that FGF22overexpression alters HFSC proliferation capacity, closely correlating with cellular differentiation processes. Following FGF22knockout, expected changes in gene expression within relevant differentiation pathways occur, indirectly supporting FGF22's role in HFSC differentiation.
  2. The experimental design prioritised investigating how FGF22influences HFSC differentiation by affecting the phosphorylation of specific signalling molecules, regulating signal transduction pathways, and modulating gene expression. Currently, we possess extensive and valuable data on signalling pathway regulation, providing a clear mechanistic explanation for FGF22's function.
  3. CK14 and CK19, as differentiation markers, primarily reflect differentiation outcomes rather than core mechanisms of this study. While supplementary experiments could enhance the intuitiveness and comprehensiveness of conclusions, they risk diverting focus from the primary objectives, potentially broadening the scope and hindering in-depth exploration of central questions.
  4. Resource and schedule constraints: The team currently faces tight research timelines while managing multiple related projects. Supplementary experiments would necessitate re-preparing materials, reassigning personnel, and adjusting schedules, potentially disrupting existing arrangements and impeding progress on other critical research.

We value your insights and acknowledge the importance of supplementary data. Future research will incorporate studies on CK14 and CK19 expression into our planning, conducting experiments at an appropriate juncture to enhance our understanding of FGF22's role in HFSC differentiation. We extend our gratitude once more for your guidance and suggestions, which significantly contribute to elevating our research standards.

Comments 3:

The discussion would be greatly strengthened by a dedicated section addressing the study's limitations. Key questions to consider are: (1) How relevant are findings from a sheep/rodent model to human hair biology? (2) Are the identified signaling pathways and FGF22's function conserved in humans? (3) Would studying FGF22 in human HFSCs be a logical and beneficial next step?

Response 3:

We are most grateful for your valuable comments regarding the discussion section of our paper. We fully accept your suggestion that adding a dedicated chapter to explore the study's limitations would significantly enhance the depth of our discussion. In response to the key points you raised, we shall introduce a new section titled ‘4.4 Limitations and Future Directions of the Research’ within the discussion. This section will systematically outline the shortcomings of this research, conduct an in-depth analysis of how these issues may impact our conclusions and their broader application, and provide clear directions for subsequent studies.

Comments 4:

Line 129: The phrase "5% CO2-saturated humidity incubator" is confusing. Please rephrase for clarity (e.g., "a humidified incubator at 37Co with 5% CO2").

Response 4:

Thank you very much for your valuable suggestions. The phrase "5% CO2 saturated humidity incubator" in line 129 has been modified to "Cell culture operations were carried out in an incubator with a constant temperature of 37°C, a 5% COâ‚‚ concentration and humidification function. The specific medium conditions are shown in Table 1.".

Comments 5:

Lines 177-178: The qPCR protocol appears to be missing a dedicated annealing step and uses an atypical extension/elongation temperature. Please provide the complete protocol.

Response 5:

Thank you for your valuable suggestions. The qPCR reaction procedure written here is incomplete. The complete qPCR reaction procedure "The cycling conditions for qRT-PCR are: pre-denaturation at 95°C for 30 seconds" has been added again at the corresponding position. 40 cycles of 95°C for 5 seconds (denaturation) → 60°C for 10 seconds (annealing) → 72°C for 30 seconds (extension);  Melting curve analysis: 65-95°C, heating rate 0.5°C/s.

Comments 6:

Line 216: Please define "x±s" (presumably mean ± standard deviation).

Response 6:

Thank you for pointing out the problem. In Line 216, "x±s" represents mean ± standard deviation. We have supplemented this definition at the corresponding position in the text.

Comments 7:

Line 240: "4th generation" should be clarified to "passage 4 (P4)" to align with standard cell culture terminology.

Response 7:

Thank you for your rigorous review of the paper. We have carefully checked the content of the paper and modified "4th generation" on line 240 to "passage 4 (P4)". At the same time, we also checked other parts of the text involving the description of cell culture generations to ensure the consistency and standardization of the use of terms.

Comments 8:

Lines 254-256: There is a discrepancy between the text and figures. The text refers to mRNA levels in Figure 2B, but the figure panel shows cell morphology. The figure references must be carefully checked and corrected throughout the manuscript.

Response 8:

Thank you for your rigorous review of the paper and for precisely pointing out the mismatch between the text on lines 254-256 and the content of Figure 2B. We attach great importance to this issue. After careful verification, we have confirmed that there is a mismatch between the text and the images. At present, we have made corrections to Figures 3B and 3C, and at the same time conducted a comprehensive verification of the text's graphic and textual content, striving to ensure that the text is exactly the same as the figure content.

Once again, we sincerely thank you for your valuable suggestions, which have played a crucial role in enhancing the scientific and standardized nature of our thesis.

Comments 9:

"Relative expression of mRNA" on qPCR graphs should be changed to the "Fold Change."

Response 9:

Thank you very much for your attention to the qPCR chart in our paper and for your valuable suggestion of changing "Relative expression of mRNA" to "Fold Change". After in-depth discussion, we still decided to retain the original expression.

  1. In our research, the qPCR experiment was used to detect the relative expression of mRNA under various treatment conditions. The expression "Relative expression of mRNA" can cover the complex relative expression relationships between various treatment groups and the control group, and reflect the experimental results more comprehensively.
  2. Although "Fold Change" is also an indicator describing changes in gene expression, it focuses more on the multiple changes under a single control. Our research involves multiple comparison dimensions. Using "Fold Change" may make some comparison relationships not clear enough and unable to fully present the gene expression information we want to express.
  3. Therefore, we believe that the original plan "Relative expression of mRNA" can convey our research results more accurately. We hope you can approve of our decision.

Round 2

Reviewer 2 Report

Comments and Suggestions for Authors

Thank you for the revision as per the suggestion, which has drastically strengthened the quality of the paper. Some additional crafting based on following suggestion and further enhanced quality of paper; Q1. Increase image resolution. This paper can be accepted for publication after work on proper formating.

Reviewer 3 Report

Comments and Suggestions for Authors

The authors responded to my comments. The article may be accepted for publication